# Visual Attentional Training Improves Reading Capabilities in Children with Dyslexia: An Eye Tracker Study During a Reading Task

**DOI:** 10.3390/brainsci10080558

**Published:** 2020-08-15

**Authors:** Simona Caldani, Christophe-Loïc Gerard, Hugo Peyre, Maria Pia Bucci

**Affiliations:** 1UMR 7114 MoDyCo, CNRS-Université Paris Nanterre, 92000 Nanterre, France; mariapia.bucci@gmail.com; 2EFEE—Centre D’Exploration Fonctionnelle de L’Équilibre Chez L’Enfant, Robert Debré Hospital, 75019 Paris, France; 3Child and Adolescent Psychiatry Department, Robert Debré Hospital, 75019 Paris, France; christophe.loic@gmail.com (C.-L.G.); peyrehugo@yahoo.fr (H.P.); 4Université de Paris, 75000 Paris, France

**Keywords:** eye movements, dyslexia, children, reading disabilities, visual attentional training

## Abstract

Dyslexia is a specific disorder in reading abilities. The aim of this study was to explore whether a short visual attentional training could improve reading capabilities in children with reading disorders by changing their oculomotor characteristics. Two groups (G1 and G2) of 25 children with reading disabilities and who are matched in IQ (intelligence quotient), sex, and age participated in the study. The allocation of a subject to a specific group (G1 = experimental group; G2 = control group) was generated in an unpredictable random sequence. The reading task was recorded twice for G1, i.e., before (T1) and after (T2) 10 min of visual attentional training. Training consisted of oculomotor tasks (saccades and pursuits movements) and searching tasks (three different exercises). For G2, the two reading tasks at T1 and T2 were done at an interval of 10 min instead. We found that at T1, oculomotor performances during reading were statistically similar for both groups of children with reading disabilities (G1 and G2). At T2, the group G1 only improved oculomotor capabilities significantly during reading; in particular, children read faster, and their fixation time was shortest. We conclude that short visual attentional training could improve the cortical mechanisms responsible for attention and reading capabilities. Further studies on a larger number of dyslexic children will be necessary in order to explore the effects of different training types on the visual attentional span given its important role on the orienting and focusing visuospatial attention and on the oculomotor performance in children with dyslexia.

## 1. Introduction

Dyslexia is a disorder reported in 5–10% of school-aged children [1]. Given the heterogeneity of this disorder in the literature there are still different theories that have tried to explain its etiology [2]. Actually, it is known that genetical investigations indicate a family risk, and the presence of a phonological impairment in dyslexia could be considered the most robust hypothesis suggesting that dyslexic children showed difficulties to read because they fail to acquire the skill of separating the sounds in a word in order to match them to their visual letter counterparts [2]. However, other hypotheses have been put forward. For instance, several studies reported deficits in auditory capabilities [3], as well as in visual perception and oculomotor performances [4,5]. Nicolson and Fawcett [6] advanced the hypothesis of a cerebellar impairment that leads to poor automaticity and motor control in children with dyslexia. Facoetti et al. [7] reported also in children with dyslexia a specific disability in orienting as well as in sustained focusing visuospatial attention. Our group further explored oculomotor capabilities in dyslexic children and suggested a deficit in visual attentional processing in relation to the immaturity of cortical structures responsible for saccades triggering [8,9,10]. Reading, in fact, corresponds to a complex cognitive process during which several mechanisms are involved: visual perception, eye movements such as saccades and fixations, and semantic and linguistic abilities; consequently, a deficit in one of these different components could cause impairment in reading acquisition. Previous studies from our group [11] and other groups [12,13] reported a longer duration of fixation in children starting to read, explained by the immaturity of cortical structures responsible for eye fixation [14]. During the normal development of reading skills, children learn to read rapidly by having shorter durations of fixations, larger prosaccades, and fewer regressions; in other words, eye movement patterns become similar to those observed in adults.

In contrast, in children with dyslexia this oculomotor pattern is not observed; indeed, it is well documented that children with dyslexia have abnormal oculomotor patterns during reading, e.g., a slower reading speed, longer duration of fixations, frequent and smaller prosaccades, and several regressions [15]. Interestingly, such characteristics have been reported in children with dyslexia of different countries speaking different languages (Greek speaking [16]; English speaking [17]; Italian speaking [18], and German speaking [19]). However, the hypothesis of an abnormal oculomotor pattern in dyslexia and related visual deficits is not shared by other researchers; for instance, the authors in [20] showed that visual deficit impairments could be causal in dyslexia, and other researchers advanced the hypothesis that poor oculomotor behavior is a consequence and not the cause of dyslexia [21]. This question is still under debate.

One of the most important objectives of research conducted on children with dyslexia is to develop training types able to improve reading skills, particularly reading speed—even if it is influenced by the type of language (more or less transparent, see previously cited studies)—as it is an important parameter for dyslexic children during school activities, given that if children need a longer time to read they cannot do the other school exercises as nondyslexic children do. Although linguistic training is the most studied [22], research has also been carried out on nonlinguistic training types.

Firstly, Solan et al. [23], in order to study some of the visual processes of reading disability, compared two types of training (eye movement and comprehension training for 12 one-hour sessions) in two groups of 15 children of 11 years old with dyslexia. For the eye movement training, the perceptual accuracy/visual efficiency method was used, while for the comprehension training the child must correct missing words in a sentence. They found that the number of fixations as well as the number of regressive saccades and comprehension were both improved by eye movement training; similar results were obtained by comprehension training, supporting the notion of a cognitive link between visual attention, eye movement’s performance, and reading comprehension. These findings support the existence of a cognitive relationship among visual attention, eye movements, and reading comprehension.

Meng et al. [24] reported reading enhancement in Chinese-speaking children with dyslexia after ten sessions within four weeks of visual perceptual training (texture discrimination tasks lasting about 50 min/session); interestingly, such improvement was still observed two months later. These authors advanced the hypothesis that temporal processing and spatial attentional capabilities leading to enlarging the visual span could be the possible cause of the training effect. This study suggested that, at least in Chinese readers, visual perceptual processing and reading ability could share a similar network.

Several studies conducted in children with dyslexia have reported an improvement of reading performance by decreasing the crowding phenomenon (i.e., increasing letter spacing [25,26,27]). It should be noted, however, that for most of these studies the number of children with dyslexia was small, and no eye movement recording during reading was performed. Our group recorded eye movements during reading and confirmed in French children with dyslexia the beneficial effect of large letter spacing between words for reading [28]. Recently, we showed also that a computer-based oculomotor training (15 min per day for five/seven days per week for 8 weeks) in a small group of 16 Italian children with dyslexia could improve reading abilities by shortening the reading time and the duration of fixation objectively measured (by an eye tracker), perhaps via visual attentional capabilities [29].

More recently, Cancer et al. [30] compared in a small group of dyslexic children (n = 12) the effect of rhythmic reading training (RRT) on reading speed and accuracy with respect to those of two training types already validated (i.e., Bakker’s visual hemisphere-specific stimulation and the action video game training). Both training types were administrated for 13 h over nine days. The authors reported that RRT improved more significantly the pseudoword reading speed, and the other training increased general reading accuracy. Authors suggested that these different results could be due to different cognitive mechanisms trained; in other words, the RTT could affect phonological awareness while the other training could have an effect on rapid automatized naming. Based on these results we could make the hypothesis that specific training types could be developed for different types of dyslexia. This hypothesis, however, needs to be tested on a large population with dyslexia.

Peters and colleagues [31] made a systematic review on the effect of visual attentional training on reading abilities in children with dyslexia aged 5–15 years old. These authors selected eighteen studies and analyzed three types of visual attentional interventions: action video games, reading acceleration, and visual perceptual training. Their findings showed that action video games improved speed reading and fluency, while reading acceleration programs increased accuracy, and speed reading and visual perceptual training benefited reading fluency and comprehension. Interestingly, these interventions can improve reading capabilities for at least two months after training. It should be noted, however, that other researchers [32] did not confirm these results. It is important also to point out that no study recorded eye movement behavior to confirm or not the eventual benefit of interventions, with the exception of one study in which a small group of children were tested (nine children only) [33].

Based on these findings, the aim of the present study is to measure by an eye tracker the effect of a short visual attentional training on the reading skills in French children with reading disorders. Eye movement recording is an objective tool that allow us to better understand the eventual changes in oculomotor patterns in children with dyslexia after training. Our final purpose is to develop a new functional training program for this kind of children.

## 2. Methods

### 2.1. Subjects

Fifty children with reading disorders (from 7.8 to 12 years old) were recruited from the Robert Debré Pediatric Hospital. The children in the study underwent a complete evaluation (see our previous studies [8,9,10,11,12,13,14,15]).

For each child, the L2MA (Langage Oral Écrit Mémoire Attention) [34] was used to measure text comprehension and the ability to read words and pseudowords. The child was included in the study if (i) their L2MA score was more than two standard deviations from the mean (all children included had dyslexia); (ii) the intelligent quotient (IQ, evaluated using the) was in normal range (between 85 and 115); (iii) they had normal visual acuity at near vision (both eyes ≥10/10); (iv) any hyperactivity deficit was excluded, using the ADHD (Attention-deficit/hyperactivity disorder) Rating Scale (ADHD-RS) parental report. It should be noted also that no treatment was given to the children, and all of them were naive of therapeutic intervention.

For each child we also evaluated the reading age by using the ELFE test (Évaluation de la Lecture en FluencE) (www.cognisciences.com, Grenoble, France) and the visual attentional (VA) span test [35]. For more details, the ELFE test is used to measure the reading age of children, and it is widely used in French laboratories/clinicians to evaluate the reading age of children.

Each child was allocated to a specific group (G1 = experimental group; G2 = control group) using an unpredictable random sequence; the group G1 has only benefited from visual attentional rehabilitation. It should be noted that the children were unaware of the experimental manipulations, and only parents were informed about the goal of the study. The investigation adhered to the principles of the Declaration of Helsinki and was accepted by the Institutional Human Experimentation Committee of CPP (Comité de protection des personnes) Ile de France I (INSERM CEEI-IRB, N° 16-290, Hotel-Dieu Hospital, Paris).

Below are details for the ELFE test and the VA (visual attention) span test. The ELFE test consisted of two texts, i.e., “Le Geant Egoiste” and “Monsieur Petit”. The child was invited to read aloud one text during 1 min, and the examiner counted the number of words read. It should be noted that the two texts are similar and comparable for their difficulty and orthography; the number of words with high and low frequency as well as the number of longer and shorter words were similar, and consequently the two texts can be used for measuring reading age before and after training without any risk of learning effect.

The VA span test consisted of 20 trials, in which the child was required to orally report a string of 5 letters that was briefly presented at the center of the monitor screen. At the beginning of each trial, a central fixation point was presented for 1000 ms followed by a blank screen for 500 ms. A letter string was then presented at the center of the display for 200 ms. The child had to report verbally all the letters immediately after they disappeared. After having written their response, the examiner pressed a button to start the next trial. At the end the examiner counted the number of letters accurately reported across the trials. Clinicians in France frequently use this test.

### 2.2. Reading Task

The text was presented on a 22-inch LCD (liquid-crystal display) screen with “full HD (High-Definition) resolution” (image size of 1920 × 1080 pixels). The child was asked to read aloud a text of four lines from a children’s book (extract from Jojo Lapin Fait des Farces, Gnid Bulton, Hachette). The paragraph contained 40 words and 174 characters. The text was 29° wide and 6.4° high; mean character width was 0.5°, and the text was written in black Courier font on a white background. This reading task is showed in Appendix A (Texts 1 and 2, see Figure A1 and Figure A2); it should be noted that only the length, subject, and writing style were balanced. Eye movements during the reading task were recorded two times at T1 and T2, i.e., before and after, respectively, 10 min of visual attentional training for the group G1 (experimental group) and also before and after 10 min for the group G2 (control group). The order of the text (Texts 1 and 2) presented in T1 and T2 was counterbalanced. It should be noted that the child was instructed to read the text without asking for comprehension.

#### 2.2.1. Visual Attentional Training

Visual attentional training consisted of both oculomotor exercises without recording eye movements (pursuits, saccades) and three searching tasks by using Metrisquare© Lebe Business Centers Sittard, Sittard, Nederland. It should be noted that the child received specific instructions for each test (see below in the “Oculomotor Task” and “Searching Task” sections).

#### 2.2.2. Oculomotor Task

Two oculomotor tasks were used: horizontal visually-guided saccades and horizontal pursuits presented on the PC (personal computer) that was used for the reading task.

For horizontal visually-guided saccades, the stimulus (both the central and eccentric targets) was a white filled circle subtending a visual angle of 0.5°. After a variable fixation period ranging between 2000 and 3500 ms, the central target disappeared, and a target at the left or at the right side of the screen simultaneously appeared for 1000 ms. The central fixation target then reappeared, signaling the beginning of the next trial. Each child performed two blocks of 30 visually-guided saccades randomly presented to the left and to the right side of different amplitudes (5°, 10°, 15°, and 20°). The child was instructed to look at the target as accurately and as rapidly as possible.

The pursuit task requires the child to follow a slowly moving visual target displayed on the PC. The target velocity was of 15°/s. The target (a white circle of 0.5°) was initially placed in the central position (0°) and then moved horizontally to one side until it reached a ±20° location, where it reversed abruptly and moved to the opposite side. A total of nine cycles were run. Children were instructed to keep their eyes on the target, wherever it moved. Children had to perform the same test four times. The duration of the pursuit task was 2 min.

#### 2.2.3. Searching Task

Three different types of searching exercises were used. The child had to search for some small objects and to remove them with a pencil following the instructions of the experimenter. The objects were presented on a tablet that was connected to a PC (Metrisquare©), and the pencil was also connected with this PC, allowing for an objective measurement of the time needed to execute the exercise, the errors, and the omission of each exercise (for more details, see Chatard et al. [36]). The three exercises are referred to as “house”, “cat”, and “space rockets”.

In the house exercise, 36 colored houses on a white background of (2 × 2 cm) were presented, among which 24 houses are on fire. The child was asked to remove the houses that are not on fire (see Figure 1A). In the cat exercise, 24 black heads of cats on a white background were presented together with 24 black stars and 24 black trees (0.5 × 1 cm). The child was asked to remove all the heads of the cats (see Figure 1B). In the space rockets exercise, 104 black space rockets having different characteristics were presented on a white background (2 × 1 cm). The child was requested to remove 16 rockets that match the model (see Figure 1C).

The duration of three searching exercises was variable (from 5 to 7 min), depending on the time needed for the child to perform the tasks. Any feedback on the performance was given to the child after doing the Metrisquare task.

### 2.3. Procedure

After the screening for dyslexia (done by the two coauthors, Dr. C.-L.G. and H.P.), the child was invited to participate in our experiment. Firstly, the examiners (two coauthors S.C. and M.P.B.) measured their reading age by the ELFE test; children were assigned randomly to each group (G1 or G2) in accordance to their performances in the VA and ELFE tests. Afterwards, the child underwent oculomotor recording (T1), and then only children of G1 (experimental group) underwent visual attentional training (i.e., both oculomotor types—without eye movements recording—and searching task by using the Metrisquare apparatus). In contrast, children of G2 (control group) did not perform the training, but they spoke for about 10 min with the two examiners. Then, T2 oculomotor recording was run for both groups of children (G1 and G2). The duration of all these tests was about 45 min because some breaks were done to avoid fatigue.

#### Eye Movement Recording

During the execution of reading task, eye movements were recorded using an Eye Brain T2^®^ (SuriCog, Paris, France) head-mounted eye tracker. This eye tracker is a medical EC (European Commission) certified device. The accuracy of this system is 0.25°, and its recording frequency reaches 300 Hz. This device can record the position of the eyes horizontally and vertically, independently and simultaneously for each eye. Calibration is done before eye movement recording. The calibration consists of a succession of red dots (diameter of 0.5°), presented on a flat PC screen of dimensions 512 × 288 mm, corresponding to the nominal diagonal size of 22 inches. During this procedure, we asked the children to look a grid of 13 points mapping the screen at a distance of 60 cm. Calibration is calculated for a 250 ms fixation period for each point. There is no obstruction of the visual field during registration with the recording system, and the calibrated zone covers a visual angle of ±22° [35]. After the calibration procedure, the reading task was explained to the child. Duration of each task varied accordingly to the specific child’s reading speed.

### 2.4. Data Analysis

We used the software MeyeAnalysis (provided with the eye tracker, SuriCog, Paris, France) in order to calibrate and to extract saccadic eye movements from the data. The onset and the end of each saccade was automatically determined through a built-in saccade detection algorithm [37]. The number and the amplitude of saccades or prosaccades, the number of regressions, the duration of fixation, and the total duration of reading were calculated in a computerized manner for each reading task. Accuracy was not measured, given that the present study focuses on eye movement pattern changes before and after visual attentional training.

### 2.5. Statistical Analysis

Statistical analysis using the GLM (general linear models) in STATISTICA® (12.0, Palo Alto, California, United States) was performed. Univariate one-way ANOVA was performed in order to compare the age, the IQ, and the number of words read in 1 min in the ELFE test and the visual attentional span test (the number of letters accurately reported across the trials) in two groups of children (G1 and G2). Repeated measurements of ANOVA were also run between the two children groups (G1 and G2) on the oculomotor parameters recorded at T1 and T2. An analysis of covariance (ANCOVA) on the duration of fixation with the total reading time as a covariable was also performed, in order to eliminate variance due to reading time differences. Post hoc comparisons were made with the Bonferroni test. The effect of a factor was considered significant when the *p*-value was below 0.05.

## 3. Results

Table 1 shows clinical characteristics of the two groups of children with reading disorders (G1 and G2). The one-way ANOVA failed to report any statistical difference between G1 (experimental group) and G2 (control group) (F_(1,48)_ = 0.15, *p* = 0.7; F_(1,48)_ = 0.10, *p* = 0.8; F_(1,48)_ = 0.02, *p* = 1.0; F_(1,48)_ = 0.01, *p* = 0.9; F_(1,48)_ = 0.03, *p* = 1.0; F_(1,48)_ = 0.04, *p* = 0.8; and F_(1,48)_ = 0.84, *p* = 0.4; respectively for the age, IQ, L2MA (oral and written languages and memory), ELFE test, and VA span test).

The ANOVA reported a significant training effect for the total reading time (F_(1,48)_ = 4.78, β = 0.09 *p* < 0.03) and also a significant interaction T × G effect (F_(1,48)_ = 39.56, β = 0.45 *p* < 0.0001). The Bonferroni post hoc test showed only that the time of reading for G1 (experimental group) at T2 decreased with respect to T1 (*p* < 0.0001) (see Figure 2).

The ANOVA showed a significant training effect for the duration of fixation (F_(1,48)_ = 6.43, β = 0.11 *p* < 0.01) and a significant interaction T × G effect (F_(1,48)_ = 17.82, β = 0.27 *p* < 0.0001). The Bonferroni post hoc test showed that for G1 (experimental group) only, the duration of fixations at T2 was shorter in contrast to T1 (*p* < 0.0001). ANCOVA showed a significant group effect in T2 (F_(1,48)_ = 10.70 *p* < 0.001), suggesting that all significant findings survived the removal of variance due to reading time differences (see Figure 3).

Table 2 shows the values of amplitude of prosaccades and regressions at T1 and T2 for both groups of children (G1 and G2). ANOVA failed to show any significant group or time effect for either interaction.

Therefore, in order to explore the eventual effect of the text in Table 3, the total time of reading and the duration of fixation are reported for children of G1 (experimental group) who read Text 1 and Text 2 before and after training (at T1 and T2, respectively). ANOVA failed to show any significant text effect, suggesting no effect of the type of the text on reading performance.

Finally, ANOVA reported only a significant interaction T × G effect for the number of regressions (F_(1,48)_ = 6.02, β = 0.11 *p* < 0.02), as seen in Figure 4. The Bonferroni test failed to show any statistical difference between the groups and the time of testing (T1 and T2).

## 4. Discussion

The aim of this study was to evaluate the effect of a short visual attentional training on the reading performance in French children with reading disabilities. Our results indicate that these children could benefit from a short visual attentional training for reading faster with a decrease in the duration of fixations. It should be noted, however, that such short visual attentional training did not lead to any change concerning the number and the amplitude of prosaccades and regressions. These findings are discussed below.

In the literature, the beneficial effects of visual perceptual training on the reading comprehension and fluency in children with dyslexia were recently reviewed (see the review of Peters et al. [31] described in the Introduction). However, that in this review, only one study [33], recorded eye movements during reading of the text. In more details, Judica and collaborators evaluated reading skills after a reading training program (1 h, two times for week for 5 months) in a small group of nine children with dyslexia (age mean of 11 years old). Training consisted of both reading a word aloud or silently and writing it on the keyboard. Eye movements were recorded using an infrared pupil reflection system with recording frequency at 200 Hz, and the duration of fixation, the number, and the amplitude of pro saccades and regressions were measured. The authors reported after training a significant shortening of fixations, suggesting that training could facilitate the process of extracting visual information on the words. In other terms, dyslexic children became more efficient in extrapolating information from the unit of letters composing the word. Indeed, during the reading task the most important eye movement parameter is the duration of fixation, given that it is during this time that the reading process takes place.

These findings suggested that visual attentional training could act on the neural network responsible of duration of fixations during reading task. In other words, the capability of orienting and focusing visuospatial attention could be improved by a visual attentional training type even if it is short in time.

We recall that neuroimaging studies reported that a complex neural network is activated during reading, predominantly the left-hemisphere area of inferior frontal, temporoparietal, and occipitotemporal cortical regions [38]. Concerning attentional system, research found that it is associated to the cingulo-frontoparietal network, linked with frontostriatal and frontoparietal pathways [39]. In the literature, dyslexia has been associated to a dysfunctional connectivity of attentional networks as well as frontal and parietal regions [40]. These authors compared the resting state using fMRI in a group of 33 children with dyslexia (age range = 7.7–15.7 years) and in a group of control children (*n* = 11), and they found in children with dyslexia a reduced activity of left intraparietal sulcus and left middle frontal gyrus; interestingly, they reported that the lower activity of these circuits was also associated with higher ratings of inattention.

We could suggest that a visual attentional training could play an important role in the cerebral networks responsible of both eye movement control and attentional mechanisms, given that attention and eye movements are linked [41]. However, attentional tests could be introduced to further explore the role of attention abilities in dyslexic children to improve their reading capabilities.

It should be noted, however, that such short visual attentional training did not lead to any change concerning the number and the amplitude of prosaccades and regressions. Judica et al. [33] reported similar findings, and they suggested that the lack of change in the number and the amplitude of saccades could be explained by the fact that the dyslexic child was not able to acquired information from the word as a single unit, and they still need to split the words. As discussed by Reichle and Sheridan [42], reading skills develop with age, and fixation duration is an important parameter for the reading capabilities. Interestingly, other studies in adult nondyslexic subjects showed improvement in reading capabilities by decreasing fixation durations and the number of saccades, in cases of a manipulation of text difficulty or text repetition [43] or shorter fixation durations when adult nondyslexic subjects had a text with larger spaces between the letters, but they failed to report any change in the number of fixations, in the total reading time, and in the comprehension scores [44]. Further studies on such issues will be necessary in order to better understand the relationship between the fixation and the saccade systems and their role in reading skills.

We suggested that such an oculomotor pattern of children with dyslexia that consists of smaller and more frequent prosaccades and regressions could be explained by the children’s reduced visual attentional span. The visual attentional span corresponded to the number of distinct visual elements that can be processed [35], and in the literature, it was reported that the orienting and focusing visuospatial attention is impaired in dyslexia [45]. More recently, Chen and colleagues [46], using a pathway analysis, studied the impact of visual attentional span in a group of 105 subjects with dyslexia (mean age 17 years old), and they observed that visual attentional span have an important impact on the reading comprehension, underlining the pivotal role of visuospatial pathways for reading processes. A reduced visual attentional span, in fact, did not permit the subjects to skip small words during reading as found in expert readers; after visual attentional training, dyslexic children are still reading as inexpert readers, and the pattern of the number and amplitude of saccades is not affected by our training type. Actually, in our coming studies we try to perform tests using visual exercises to increase the visual attention span in dyslexic children in order to explore possible improvement also in the number and amplitude of saccades during reading, as it has been recently showed by Zhao et al. [47].

## 5. Limitations

The present study has several limitations that are worth noting. First, we did not examine the psycholinguistic effect of the texts used, and the type of font is known to be quite complex for dyslexic children. Secondly, the paragraph that the child has to read was quite short given the technical set up of eye movement recordings, particularly due to the calibration of the eye tracker system. Indeed, to avoid head movements of the child during the reading task, a short paragraph was presented in the center of the screen, and reading accuracy was not recorded.

Another important factor needing to be taken in account is the motivation of children; children of the control group may have been less motivated with respect to children of the experimental group. Further studies taking in account these arguments need to be done on a large group of dyslexic children by using a more ecological experimental setup.

## 6. Conclusions

A short visual attentional training could improve the cortical mechanism responsible for focus attention and reading capabilities in children with dyslexia. Further studies will be necessary in order to test different training types for improving both visual attentional span and eye movements during reading in a dyslexic population.

## Figures and Tables

**Figure 1 brainsci-10-00558-f001:**
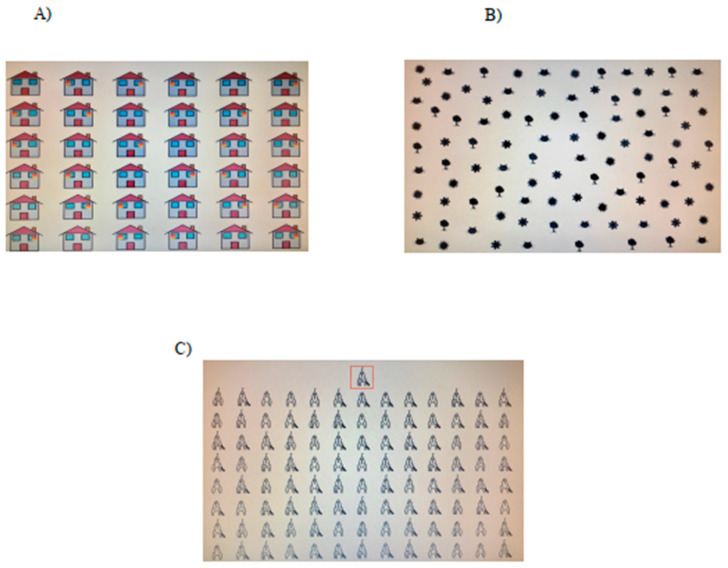
Visual attentional exercises by Metrisquare©: house (**A**), cats (**B**), and space rockets (**C**).

**Figure 2 brainsci-10-00558-f002:**
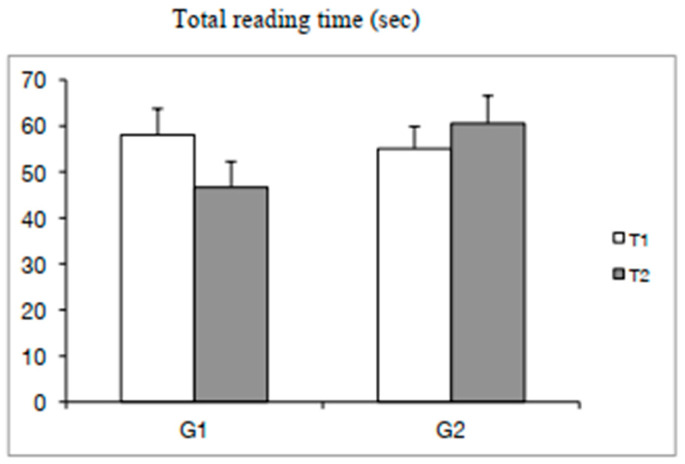
Means and standard deviations of the total time (s) of reading the text at T1 and T2 for both groups of children (G1 = experimental group; G2 = control group).

**Figure 3 brainsci-10-00558-f003:**
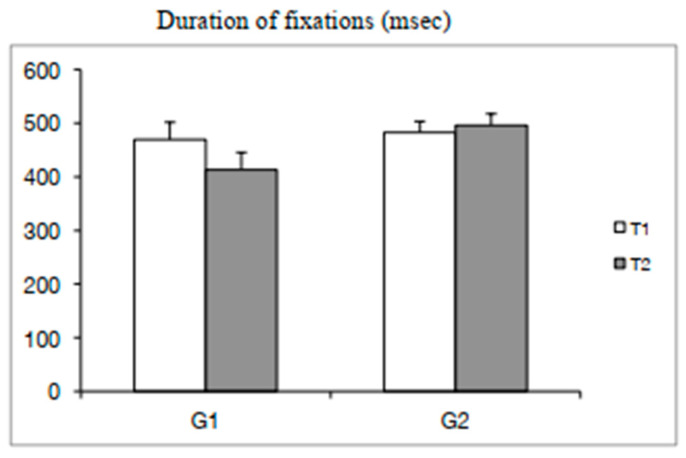
Means and standard deviations of the duration of fixations (ms) at T1 and T2 for both groups of children (G1 = experimental group; G2 = control group).

**Figure 4 brainsci-10-00558-f004:**
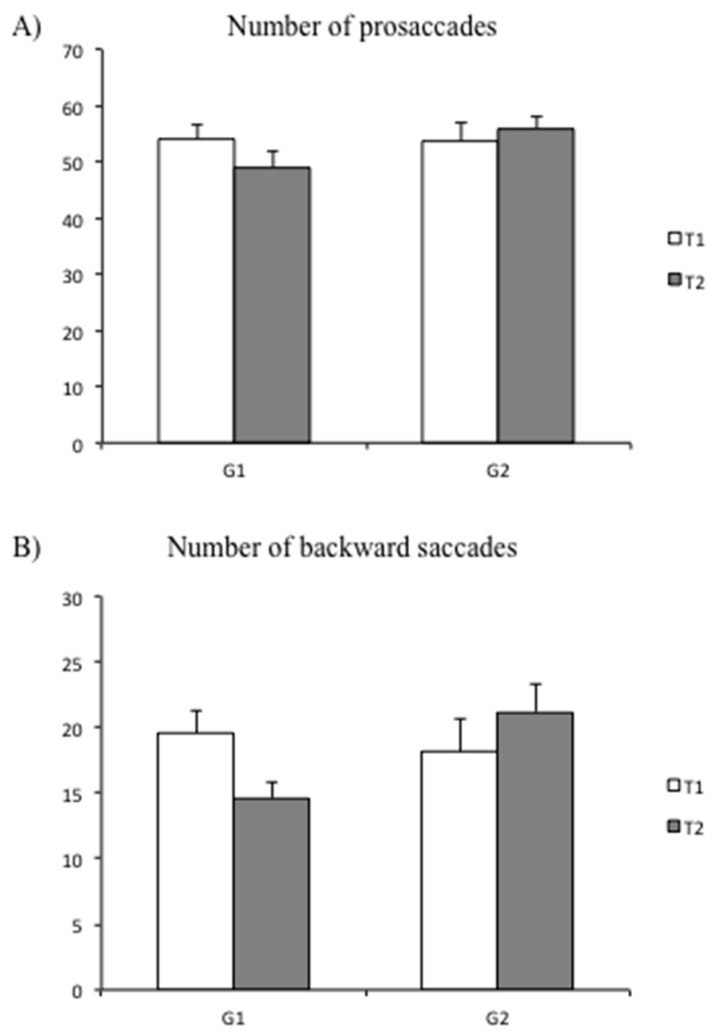
Means and standard deviations of the number of prosaccades (**A**) and number of regressions (**B**) at T1 and T2 for both groups of children (G1 = experimental group; G2 = control group).

**Table 1 brainsci-10-00558-t001:** Clinical characteristics of the two groups of dyslexic children (G1 = experimental group; G2 = control group) with mean and standard deviations of age (years), IQ, of words/min read in the ELFE test and visual attentional span test (the number of letters accurately reported across the trials).

	G1 (*n* = 25)	G2 (*n* = 25)
Age (years)	9.56 ± 0.29	9.74 ± 0.38
IQ (WISC-IV)	100 ± 6	102 ± 5.1
ADHD-RS score	5.2 ± 1	4.9 ± 1.2
L2MA standard deviation from the mean		
Oral Language	2.8	2.9
Written Language	2.6	2.7
Memory	2.7	2.8
ELFE test	48 ± 5.3	50 ± 5.8
VA span	63 ± 3.0	60 ± 2.1

Note: For the L2MA test done in both group of children the standard deviation from normal mean is reported.

**Table 2 brainsci-10-00558-t002:** Means and standard deviations of the amplitude of prosaccades and regressions at T1 and T2 for both groups of children (G1 = experimental group; G2 = control group).

	Amplitude of Prosaccades (°)		Amplitude of Regressions (°)	
	T1	T2	T1	T2
G1 (experimental group)	2.7 ± 1.1	2.5 ± 0.1	2.6 ± 0.1	2.6 ± 0.1
G2 (control group)	2.5 ± 0.1	2.4 ± 0.1	2.7 ± 0.1	2.6 ± 0.1

**Table 3 brainsci-10-00558-t003:** Mean and standard deviations of the total time of reading and of duration of fixation for children of G1 = experimental group, who read Texts 1 and 2 before and after training (in T1 and T2).

Total Time of Reading (s)				Duration of Fixation (ms)			
Before Training (T1)		After Training (T2)		Before Training (T1)		After Training (T2)	
Text 1	Text 2	Text 1	Text 2	Text 1	Text 2	Text 1	Text 2
59 ± 5	57 ± 6	45 ± 6	47 ± 5	465 ± 25	475 ± 29	411 ± 21	415 ± 20

## Data Availability

The datasets analyzed during the current study are available from the corresponding author on reasonable request.

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
