# Peer review of "Visual Attentional Training Improves Reading Capabilities in Children with Dyslexia: An Eye Tracker Study During a Reading Task"

_brainsci, 2020, doi:10.3390/brainsci10080558_

Round 1

Reviewer 1 Report

The revisions in response to my comments are not clearly highlighted in the manuscript - therefore it was difficult for me to find them. Also, the numbers of the pages that the authors refer to in the comments do not correspond to the actual manuscript.

Abstract

-"achieve automaticity in reading and writing." Dyslexia is a specific disorder in reading abilities. Although it is often associated with writing difficulties, the latter are not listed in the inclusion criteria for the diagnosis of Dyslexia.

Introduction

-The authors claimed "It is important also to point out that any study recorded eye movements’ behavior to confirm or not eventual benefit of interventions.". However, in the Discussion they wrote that "only one study, recorded eye movements during reading text." citing Judica et al. work.

Methods

-"Note also that any drug treatment has been given to children": do authors mean "No treatment was given to children"? Why was it important to specify that, if children had no major comorbidities with other disorders?

-I would not say that "all French laboratory/clinicians used the ELFE test"; I suggest to rather explain that it is widely used in France.

-Authors described extensively the ELFE and VA tests in the subject section; I think it is better to move these descriptions in later paragraphs.

-Authors stated that the reading tasks in T1 and T2 had "similar words characteristics"; However, they did not compare word characteristics - beside the texts being extracted from the same book, and therefore they were written by the same author. I suggest to change this sentence, specifying that only length, subject and writing style were balanced. Furthermore, in the discussion section they wrote: "we did not examine psycholinguistic indices of the texts.". I think it is not clear what they are referring to (what indexes? for what reason were they suppose to analyse them?)

-"or something else that could stress the child" I think this is a vague statement (what could the researcher ask to the child to induce stress, which was not asked?)

-Authors explained that "Each child was allocated to a specific group (G1 or G2) using an unpredictable random sequence" but later on they claimed that "children were paired in the two groups (G1 or G2)." I suppose they mean they were assigned to each group in accordance to their performances in the VA and ELFE tests. Did authors use random or stratified assignment? Please, explain.

Discussion

-"reading skills, particularly reading speed, that, even if it is influenced by the type of language (more or less transparent), it is an important parameter for dyslexic children during school activities." I think authors should provide previous evidence to support this claim and they should better address the decision of including only speed as the reading outcome.

Author Response

Reviewer 1:

Abstract

-"achieve automaticity in reading and writing." Dyslexia is a specific disorder in reading abilities. Although it is often associated with writing difficulties, the latter are not listed in the inclusion criteria for the diagnosis of Dyslexia.

This sentence was corrected according to your suggestion

Introduction

-The authors claimed "It is important also to point out that any study recorded eye movements’ behavior to confirm or not eventual benefit of interventions.". However, in the Discussion they wrote that "only one study, recorded eye movements during reading text." citing Judica et al. work.

We corrected this sentence in the Introduction. Note that the study of Judica was conducted on 9 children only.

Methods

-"Note also that any drug treatment has been given to children": do authors mean "No treatment was given to children"? Why was it important to specify that, if children had no major comorbidities with other disorders?

Yes we changed the sentence. A reviewer asked to specify that

-I would not say that "all French laboratory/clinicians used the ELFE test"; I suggest to rather explain that it is widely used in France.

The sentence was corrected

-Authors described extensively the ELFE and VA tests in the subject section; I think it is better to move these descriptions in later paragraphs.

The description of the two tests was moved at the end of the subject section as you suggested

-Authors stated that the reading tasks in T1 and T2 had "similar words characteristics"; However, they did not compare word characteristics - beside the texts being extracted from the same book, and therefore they were written by the same author. I suggest to change this sentence, specifying that only length, subject and writing style were balanced. Furthermore, in the discussion section they wrote: "we did not examine psycholinguistic indices of the texts.". I think it is not clear what they are referring to (what indexes? for what reason were they suppose to analyse them?)

The first sentence was changed as you suggested. In the Limitation section a previous reviewer asked to insert the word ‘indice’; now we changed this word in ‘effect’

-"or something else that could stress the child" I think this is a vague statement (what could the researcher ask to the child to induce stress, which was not asked?)

This sentence was eliminated

-Authors explained that "Each child was allocated to a specific group (G1 or G2) using an unpredictable random sequence" but later on they claimed that "children were paired in the two groups (G1 or G2)." I suppose they mean they were assigned to each group in accordance to their performances in the VA and ELFE tests. Did authors use random or stratified assignment? Please, explain.

This sentence was corrected (see page 13)

Discussion

-"reading skills, particularly reading speed, that, even if it is influenced by the type of language (more or less transparent), it is an important parameter for dyslexic children during school activities." I think authors should provide previous evidence to support this claim and they should better address the decision of including only speed as the reading outcome.

This point was better explained (see page 6). The importance of duration fixation is more discussed also (see pages 18-19)

Reviewer 2 Report

Review of ms “Visual attentional training improves reading capabilities in children with dyslexia: an eye tracker study during a reading task”

The authors reported a pre-post experiment with children with dyslexia in which half of the children had participated in a short (10 minute) visual training program. The authors found faster reading times post-treatment than pre-treatment for the experimental group but not for the control group.

Overall, and despite its limitations (see below), the findings presented here are interesting and suggestive, so I am quite positive. Here is my list of comments:

--When discussing the facilitation from extra space (line 95) it’s of course ok to cite Zorzi et al. (2012), but Perea et al. (2012; doi: 10.1016/j.learninstruc.2012.04.001) were the first to report this effect (i.e., the Perea et al. paper was already published before the Zorzi et al. paper was submitted for publication) so the two papers should be cited. Also, a parenthesis was missing on line 95.

--G1 should be renamed “Experimental group” and G2 should be renamed as “Control group).

--The way the ANOVAs were reported was a bit awkward. I would report first the main effects, the interaction. If the interaction is significant, then the authors may conduct the simple tests effects. Also, I would not make a constant call to the Figures.

--The authors should examine in greater detail the fact that the benefit in reading times in the experimental groups was due to shorter fixation durations (the “when” in eye movement control) and not to the number of fixations (either pro-saccade or regressive saccades; the “where” in eye movement control). In many scenarios, one gets parallel effects (i.e., briefer fixation durations and fewer saccades, in case of a manipulation of text difficulty or text repetition; for example: doi:10.1037/1196-1961.49.2.151) or the opposite effects (i.e., briefer fixation durations but more saccades, as in the case of inter-letter spacing; e.g., doi:10.1017/sjp.2016.28). I believe the authors should link their findings with models of eye movement control during reading (e.g., EZ-Reader model; see for instance 10.1093/oxfordhb/9780199324576.013.17).

--To me, the limitation of the experiment is that the lack of a “control task” with no relevant attentional involvement to check a null effect across groups. The issue now is that perhaps the children in the control (G2) group were less motivated than the children in the experimental group. This is something that needs to be indicated in the Discussion—and this is something that the authors may want to consider in future work (perhaps the authors may want to stress the exploratory nature of this work).

--The Discussion section is at times a bit too speculative, as one would need to have manipulated another factor to examine in greater depth to what degree attention was involved in the observed pattern.

--The authors should check for typos and grammar in a revised version of the manuscript.

Author Response

Reviewer 2:

The authors reported a pre-post experiment with children with dyslexia in which half of the children had participated in a short (10 minute) visual training program. The authors found faster reading times post-treatment than pre-treatment for the experimental group but not for the control group.

Overall, and despite its limitations (see below), the findings presented here are interesting and suggestive, so I am quite positive. Here is my list of comments:

- When discussing the facilitation from extra space (line 95) it’s of course ok to cite Zorzi et al. (2012), but Perea et al. (2012; doi: 10.1016/j.learninstruc.2012.04.001) were the first to report this effect (i.e., the Perea et al. paper was already published before the Zorzi et al. paper was submitted for publication) so the two papers should be cited. Also, a parenthesis was missing on line 95.

We agree with you, the Perea paper was added as well as the parenthesis.

- G1 should be renamed “Experimental group” and G2 should be renamed as “Control group).

We added the name you suggested in the text.

--The way the ANOVAs were reported was a bit awkward. I would report first the main effects, the interaction. If the interaction is significant, then the authors may conduct the simple tests effects. Also, I would not make a constant call to the Figures.

Result section improved following your suggestions

--The authors should examine in greater detail the fact that the benefit in reading times in the experimental groups was due to shorter fixation durations (the “when” in eye movement control) and not to the number of fixations (either pro-saccade or regressive saccades; the “where” in eye movement control). In many scenarios, one gets parallel effects (i.e., briefer fixation durations and fewer saccades, in case of a manipulation of text difficulty or text repetition; for example: doi:10.1037/1196-1961.49.2.151) or the opposite effects (i.e., briefer fixation durations but more saccades, as in the case of inter-letter spacing; e.g., doi:10.1017/sjp.2016.28). I believe the authors should link their findings with models of eye movement control during reading (e.g., EZ-Reader model; see for instance 10.1093/oxfordhb/9780199324576.013.17).

The references you suggested were cited in the Discussion (see pages 18-19)

--To me, the limitation of the experiment is that the lack of a “control task” with no relevant attentional involvement to check a null effect across groups. The issue now is that perhaps the children in the control (G2) group were less motivated than the children in the experimental group. This is something that needs to be indicated in the Discussion—and this is something that the authors may want to consider in future work (perhaps the authors may want to stress the exploratory nature of this work).

Limitation section improved following your suggestion (see page 20)

--The Discussion section is at times a bit too speculative, as one would need to have manipulated another factor to examine in greater depth to what degree attention was involved in the observed pattern.

This point was included in the Discussion (see page 18)

--The authors should check for typos and grammar in a revised version of the manuscript.

Complied

Round 2

Reviewer 2 Report

I think the ms did a good job in this revised version of the ms. I just have a minor edit

line 352

by using more ecological experimental set up.

should be 

by using more ecological experimental setups

in plural or adding an “a” as

in a more ecological experimental set up.

This manuscript is a resubmission of an earlier submission. The following is a list of the peer review reports and author responses from that submission.

Round 1

Reviewer 1 Report

Title:

-The authors should clearly explain if by “children with reading disabilities” they refer to children with a diagnosis of dyslexia, or else.

Abstract:

-I suggest to change the term 'literacy' - which is, more broadly, the ability to read and write. Children with dyslexia do learn how to read and write, however, their reading process is not automatised.

Introduction:

-To my knowledge, hereditariness of dyslexia has not been proven. Genetical investigations indicated solely a family risk.

-I suggest the authors could include a more recent evidence about the efficacy of the Action Video Game (AVG) Training in combination with visual hemispheric specific stimulation (VHSS), which is not included in Peters et al.’s (2019) systematic review:

Cancer, A., Bonacina, S., Antonietti, A., Salandi, A., Molteni, M., & Lorusso, M. L. (2020). The Effectiveness of Interventions for Developmental Dyslexia: Rhythmic Reading Training Compared With Hemisphere-Specific Stimulation and Action Video Games. Frontiers in Psychology, 11. https://doi.org/10.3389/fpsyg.2020.01158.

-I suggest that the authors should also address the study by Łuniewska et al. (2018) which found no difference between reading improvements after AVG training and the spontaneous development of reading abilities – as measured in a no-treatment control group in a Polish population.

Łuniewska, M., Chyl, K., Dębska, A., Kacprzak, A., Plewko, J., Szczerbiński, M., et al. (2018). Neither action nor phonological video games make dyslexic children read better. Scientific Reports 8, 549. doi:10.1038/s41598-017-18878-7.

-line 61: I suggest to further explained why reading speed is the preferred target for intervention. This is related to the depth of the orthography, which varies in different languages (e.g., dyslexia affects mostly reading speed in transparent languages, but this is not the case in more deep orthographies, such as English).

Methods:

-Authors should further address the inclusion criteria for participants: Did all participant received a diagnosis of dyslexia prior to the recruitment to the investigation? If not, were they on a waiting list for a cognitive/learning ability assessment? What about comorbilities with other disorders (e.g., ADHD)?

-Authors should clearly declare their hypothesis, and report which they consider as the primary outcome of their intervention - eye-movements or reading performance? Or both?

-I suggest the authors should provide more information about the characteristics of the ELFE and VA span tasks.

-In the method section, a Procedure paragraph should be included to help the reader discriminating between the assessment tasks from the training tasks. Furthermore, information about the assessment and training settings should be reported (who administered the tasks and the training? When were the ELFE and visual span tests administered? In the same day or else?)

-Authors should clearly explain that they applied the ELSE and the visual task only in T1 to test that the two groups’ abilities were paired at baseline.

-Why was the Courier font used for the reading task? Did the authors have specific hypothesis about the characteristics of the font? Courier is a serif typeface, and it is known that children with dyslexia are facilitated by sans serif fonts (such as Arial), as compared to serif ones, because less visual information has to be processed.

-The reading task was administer in T2 ‘using a different test with similar word characteristics’. Which  psycholinguistic indexes have the authors considered? Did they compare them quantitatively between the texts? Did they counterbalanced the order of the texts presented in T1 and T2?

-Authors should mention they recorded reading speed in the computerised reading task and they should also include why they considered reading speed only, and not accuracy.

-In the Metrisquare visual search task, did the children receive a feedback of their performance? I would expect a training task to include an online error signal.

Limitations:

-I wouldn’t list the lack of reading comprehension measures, since this ability is only partially explained by reading abilities.

Minor language issues:

-You sometimes use ‘visual attentional’ and some other times ‘visuo-attentional’. You should uniform that.

-line 121 ‘Participants were ask to read’ is a repetition of the previous sentence ‘Child was asked to read”.

Author Response

Reviewer 1:

Title:

-The authors should clearly explain if by “children with reading disabilities” they refer to children with a diagnosis of dyslexia, or else.

Complied

Abstract:

-I suggest to change the term 'literacy' - which is, more broadly, the ability to read and write. Children with dyslexia do learn how to read and write, however, their reading process is not automatised.

Complied

Introduction:

-To my knowledge, hereditariness of dyslexia has not been proven. Genetical investigations indicated solely a family risk.

Complied

-I suggest the authors could include a more recent evidence about the efficacy of the Action Video Game (AVG) Training in combination with visual hemispheric specific stimulation (VHSS), which is not included in Peters et al.’s (2019) systematic review:

Cancer, A., Bonacina, S., Antonietti, A., Salandi, A., Molteni, M., & Lorusso, M. L. (2020). The Effectiveness of Interventions for Developmental Dyslexia: Rhythmic Reading Training Compared With Hemisphere-Specific Stimulation and Action Video Games. Frontiers in Psychology, 11. https://doi.org/10.3389/fpsyg.2020.01158.

-I suggest that the authors should also address the study by Łuniewska et al. (2018) which found no difference between reading improvements after AVG training and the spontaneous development of reading abilities – as measured in a no-treatment control group in a Polish population.

Łuniewska, M., Chyl, K., Dębska, A., Kacprzak, A., Plewko, J., Szczerbiński, M., et al. (2018). Neither action nor phonological video games make dyslexic children read better. Scientific Reports 8, 549. doi:10.1038/s41598-017-18878-7.

The two references has been added in the Introduction (pages 6 and 7)

-line 61: I suggest to further explained why reading speed is the preferred target for intervention. This is related to the depth of the orthography, which varies in different languages (e.g., dyslexia affects mostly reading speed in transparent languages, but this is not the case in more deep orthographies, such as English).

This point has been explained (page 5)

Methods:

-Authors should further address the inclusion criteria for participants: Did all participant received a diagnosis of dyslexia prior to the recruitment to the investigation? If not, were they on a waiting list for a cognitive/learning ability assessment? What about comorbilities with other disorders (e.g., ADHD)?

The description on the selection of patients improved (see page 8)

-Authors should clearly declare their hypothesis, and report which they consider as the primary outcome of their intervention - eye-movements or reading performance? Or both?

We think that eye movements performance could give an objective information on eventual benefit of training on reading performance given that the eye movements patterns during reading is well known in dyslexic as well as non dyslexic children (see several studies in Greek speaking12; English speaking13; Italian speaking,14 and German speaking15 and in French population from our group).

-I suggest the authors should provide more information about the characteristics of the ELFE and VA span tasks.

ELEF test has been developed by the laboratory Cogniscience in Grenoble University and the Ministry of Education. It is actually the test used to detect the reading age of children and all French laboratory/clinicians used this test to evaluate reading age of child. There are two tests for reading (‘Le Geant Egoiste’ and ‘Monsieur Petit’). Child is invited to read aloud the text during 1 minute. Then the examiner count the number of words read (information has been added in page 8).

VA span has been introduced by Valdois’s group (Bosse et al. 2007). This test is now explained (see page 9).

-In the method section, a Procedure paragraph should be included to help the reader discriminating between the assessment tasks from the training tasks. Furthermore, information about the assessment and training settings should be reported (who administered the tasks and the training? When were the ELFE and visual span tests administered? In the same day or else?)

Procedure paragraph has been added; we hope that the experimental procedure is more clear (see pages 11-12).

-Authors should clearly explain that they applied the ELFE and the visual task only in T1 to test that the two groups’ abilities were paired at baseline.

Complied, see Procedure paragraph (page 11-12)

-Why was the Courier font used for the reading task? Did the authors have specific hypothesis about the characteristics of the font? Courier is a serif typeface, and it is known that children with dyslexia are facilitated by sans serif fonts (such as Arial), as compared to serif ones, because less visual information has to be processed.

We agree with you, however the goal of the study was not to test the effect of different font types. Previous studies done by our group (Bucci et al, 2012; Seassau et la. 2014; Seassau & Bucci, 2013) testing reading capabilities in dyslexic as well as non dyslexic children already used this font. We decided to use the same font of our previous works in order to have possible comparison. This point was added in the Limitation paragraph.

-The reading task was administer in T2 ‘using a different test with similar word characteristics’. Which  psycholinguistic indexes have the authors considered? Did they compare them quantitatively between the texts? Did they counterbalanced the order of the texts presented in T1 and T2?

The text (Text 1 and text 2) to read at T1 and T2 was paragraph contained 40 words and 174 characters and both were extracted from children’s book (“Jojo Lapin fait des farces”, Gnid Bulton, Hachette); the order of the texts presentation was counterbalanced. We did not examine other psycholinguistic indices (this point was added in the Limitation paragraph). The two texts had been added in appendix

-Authors should mention they recorded reading speed in the computerised reading task and they should also include why they considered reading speed only, and not accuracy.

This was added in page 9.

-In the Metrisquare visual search task, did the children receive a feedback of their performance? I would expect a training task to include an online error signal.

No, any feedback of their performance was given to children after Metrisquare task (see page 11)

Limitations:

-I wouldn’t list the lack of reading comprehension measures, since this ability is only partially explained by reading abilities.

This chapter improved following your suggestion and those of the Reviewer 2

Minor language issues:

-You sometimes use ‘visual attentional’ and some other times ‘visuo-attentional’. You should uniform that.

Complied

-line 121 ‘Participants were ask to read’ is a repetition of the previous sentence ‘Child was asked to read”.

The sentence was deleted

Reviewer 2 Report

  1. The experiment was straightforward and the eye-tracking measures were described well.  The paper also addresses an interesting and ongoing issue in dyslexia: how to help children with dyslexia improve reading speed. Unfortunately, this strength of the study (namely, the focus on reading speed) is sidestepped beginning with the review of the literature. A short review of why reading speed (or probably more appropriately, reading fluency) is a necessary target would help provide the reader with a context and rationale for the experiment.
  2. Dyslexia diagnoses and reading characteristics of the two groups should be described more thoroughly. Means, SDs, and ranges for both groups on reading measures used to diagnose children with dyslexia and reliability measures for these diagnostic instruments should be provided. Reading performance of developing readers (such as the participants in this study, ages 7-12) can vary widely and any number of reading subskills could greatly impact reading speed, especially on such a short task. Table 1 shows reading age from the ELFE but the authors also state participants were administered the L2MA yielding comprehension and decoding scores, yet these scores were not provided. Were groups matched on these two variables? These, in my view, are preferred to “age equivalency” measures which are known to be psychometrically problematic.
  3. Children with dyslexia (especially children learning to read deep and opaque orthographies such as French or English) are not only slow readers, but also frequently inaccurate. How did the researchers control for variability in reading accuracy? This seems especially important considering the wide age range (7-12) of the participants. Many children age 7 have not yet mastered decoding and would not be expected to read fluently a passage written for a 9-year-old. Reading speed should only be measured on passages for which a child has very high accuracy. To get a true reading speed measure, you might stipulate, for example, that passages read with more than three errors would not be included in the analyses. Alternatively, to account for reading accuracy errors, the researchers may want to consider using reading fluency (an index that combines accuracy + speed) as an outcome measure.
  4. G1 was administered a training task between reading T1 and reading T2, but G2 was not administered a control task between readings. What did they do during this time? Without an alternative task to treatment, it is not possible to confidently attribute the observed findings (increased reading speed and reduced duration of foveal fixations) directly to 10 minutes of visual attention intervention, per se. Is it possible that doing anything visually engaging could account for the observed changes in G1?
  5. The first reading passage was not adequately described and the second reading passage was not described at all. To show that they are matched, both passages should be described more thoroughly using lexical variables or other quantitative indexes. Reliability measures should also be provided. I’m especially curious about this because Figure 2 shows the mean reading speed for G2 increasing from T1 to T2. There are many variables that can be used as controls for reading passages to show that they are similar (e.g., type-token ratio, readability indices, grade level, etc.). Ideally, both reading passages should be included in an appendix. A better alternative would be to use standardized reading passages.
  6. No standard protocol was provided for administering the reading tasks. For example, were the children instructed to read “as fast as you can” or “for comprehension”? Did they read aloud or silently? This is a very important considering fixation durations are longer for oral reading than for silent reading. If participants were instructed to read aloud, how did the experimenters handle reading errors? Restarts? Was there any preface regarding the passage? Is the story well known enough that children could guess some of the words from context? Again, how does this compare with Passage 2?
  7. The explanation for why fixation duration (not saccades) was affected by the treatment was unclear and the argument seemed a bit circular.

Other issues that should be addressed:

Recruitment procedures should be outlined briefly.

In addition to F and p values, means and SDs for all four outcome measures and effect sizes or power for statistical tests should be provided.

A more thorough attempt at addressing limitations to the study should be provided.

The relation between visual skills and reading difficulties in dyslexia has a long history and is quite controversial. This was not addressed. Consider including some of this literature as well as some seminal eye-tracking studies of reading from the developmental literature.

Author Response

Reviewer 2:

  1. The experiment was straightforward and the eye-tracking measures were described well.  The paper also addresses an interesting and ongoing issue in dyslexia: how to help children with dyslexia improve reading speed. Unfortunately, this strength of the study (namely, the focus on reading speed) is sidestepped beginning with the review of the literature. A short review of why reading speed (or probably more appropriately, reading fluency) is a necessary target would help provide the reader with a context and rationale for the experiment.

The introduction improved following your suggestion (see page 5)

  1. Dyslexia diagnoses and reading characteristics of the two groups should be described more thoroughly. Means, SDs, and ranges for both groups on reading measures used to diagnose children with dyslexia and reliability measures for these diagnostic instruments should be provided. Reading performance of developing readers (such as the participants in this study, ages 7-12) can vary widely and any number of reading subskills could greatly impact reading speed, especially on such a short task. Table 1 shows reading age from the ELFE but the authors also state participants were administered the L2MA yielding comprehension and decoding scores, yet these scores were not provided. Were groups matched on these two variables? These, in my view, are preferred to “age equivalency” measures which are known to be psychometrically problematic.

In Table 1we added L2MA score

  1. Children with dyslexia (especially children learning to read deep and opaque orthographies such as French or English) are not only slow readers, but also frequently inaccurate. How did the researchers control for variability in reading accuracy? This seems especially important considering the wide age range (7-12) of the participants. Many children age 7 have not yet mastered decoding and would not be expected to read fluently a passage written for a 9-year-old. Reading speed should only be measured on passages for which a child has very high accuracy. To get a true reading speed measure, you might stipulate, for example, that passages read with more than three errors would not be included in the analyses. Alternatively, to account for reading accuracy errors, the researchers may want to consider using reading fluency (an index that combines accuracy + speed) as an outcome measure.

Reading accuracy was not measured (this was added in the Limitation section). Note, however, after the reading task few questions were asked to the child in order to be sure that they understood correctly the passage read. Even if some reading error was done, all children understood well the text.

  1. G1 was administered a training task between reading T1 and reading T2, but G2 was not administered a control task between readings. What did they do during this time? Without an alternative task to treatment, it is not possible to confidently attribute the observed findings (increased reading speed and reduced duration of foveal fixations) directly to 10 minutes of visual attention intervention, per se. Is it possible that doing anything visually engaging could account for the observed changes in G1?

Procedure section was added in order to better explain the study (see pages 11-12). Children of G2 did not engage their visual attention during the 10 minutes between T1 and T2 recordings.

  1. The first reading passage was not adequately described and the second reading passage was not described at all. To show that they are matched, both passages should be described more thoroughly using lexical variables or other quantitative indexes. Reliability measures should also be provided. I’m especially curious about this because Figure 2 shows the mean reading speed for G2 increasing from T1 to T2. There are many variables that can be used as controls for reading passages to show that they are similar (e.g., type-token ratio, readability indices, grade level, etc.). Ideally, both reading passages should be included in an appendix. A better alternative would be to use standardized reading passages.

The two texts (1 and 2) are now presented in the Appendix.

  1. No standard protocol was provided for administering the reading tasks. For example, were the children instructed to read “as fast as you can” or “for comprehension”? Did they read aloud or silently? This is a very important considering fixation durations are longer for oral reading than for silent reading. If participants were instructed to read aloud, how did the experimenters handle reading errors? Restarts? Was there any preface regarding the passage? Is the story well known enough that children could guess some of the words from context? Again, how does this compare with Passage 2?

We improved the description of the reading task (see page 9). Note that if the child made an error we did not ask to restart to read the words. We decided to avoid any intrusion from the examiner during the task. The passage was extract from a child’s book consequently child had no difficulty to read the text (see previous works form our group).

  1. The explanation for why fixation duration (not saccades) was affected by the treatment was unclear and the argument seemed a bit circular.

Complied (see page 16)

Other issues that should be addressed:

Recruitment procedures should be outlined briefly.

Complied see page 8.

In addition to F and p values, means and SDs for all four outcome measures and effect sizes or power for statistical tests should be provided.

The effect sizes is now added in the Result section (page 14).

A more thorough attempt at addressing limitations to the study should be provided.

Limitation chapter was improved (pages 17-18).

The relation between visual skills and reading difficulties in dyslexia has a long history and is quite controversial. This was not addressed. Consider including some of this literature as well as some seminal eye-tracking studies of reading from the developmental literature.

Introduction improved (page 5).